# The relationship between the neutrophil percentage to albumin ratio and the occurrence of prostate cancer

Ping Kong[☉], Lei Yang[☉], Haibing Wang[☉], Zhiqi Liu[¤], Zhiqiang Zhang[☉]*

Department of Urology, Second Affiliated Hospital of Anhui Medical University, Hefei, Anhui, China

[☉] These authors contributed equally to this study.
[¤] Current address: Department of Urology, The Second Affiliated Hospital of Anhui Medical University, No. 678 Fu-Rong Road, 230106 Hefei, Anhui, China
* zhangzq_urology@163.com

## Abstract

### Background

Although prior research has indicated that nutritional and inflammatory markers may play a role in prostate cancer development, the exact interplay and underlying mechanisms are not yet fully understood.

### Materials and Methods

The study population comprised 16,481 males from the NHANES database, after excluding participants with missing covariates. The Neutrophil Percentage-to-Albumin Ratio (NPAR) was used to assess the inflammation and nutritional status. Statistical methods such as multivariable logistic regression, eXtreme Gradient Boosting model, subgroup analysis, and Generalized Additive Model were used to analyze the relationship between NPAR and prostate cancer prevalence.

### Results

The restricted cubic splines of the independent variable NPAR and the dependent variable (prostate cancer prevalence) were statistically significant based on the logistic regression analysis. The eXtreme Gradient Boosting machine learning model identified the NPAR and age as the most influential variables for prostate cancer. Subgroup analysis revealed significant correlations between the NPAR and age, race, and smoking status. Clinical validation has confirmed the diagnostic significance of NPAR in prostate cancer.

**Data availability statement:** Publicly available datasets were analyzed in this study. This data can be found here: https://www.cdc.gov/nchs/nhanes/index.htm. Clinical data is fully anonymous in a manner. This data can be found here: https://data.mendeley.com/datasets/hyf5jp-p26f/1 (less...).

**Funding:** Precision Diagnosis and Classification of Prostate Cancer via Artificial Neural Network-Driven Intelligent Analysis of MR Images.

**Competing interests:** The authors have declared that no competing interests exist.

## Conclusion

A positive correlation was observed between NPAR and prostate cancer prevalence, indicating the potential mechanism of developing the disease. However, due to the cross-sectional design and self-reported cancer diagnoses in the NHANES database, causality cannot be established.

## Introduction

Prostate cancer (PCa) is the most common solid organ tumor among men in developed countries, second only to lung cancer, and ranks as the second most common solid organ tumor globally [1]. More than 95% of PCa cases are adenocarcinomas, with the majority originating from the acini and a minority originating from the ducts. Nearly 80% of prostate cancers arise from luminal or basal epithelial cells in the peripheral zone, which constitutes more than 70% of the total prostate tissue.The prevalence of PCa is approximately 60% among men over 65 years old; however, its pathogenesis remains unclear [1–3]. The etiology of prostate cancer is complex and involves genetic polymorphisms, family history, elevated body mass index (BMI), age, African ancestry, and smoking. However, only three risk factors for prostate cancer are currently widely recognized: age (the risk of prostate cancer increases with advancing age), race (African Americans have an elevated risk), and genetics (family history of prostate cancer, particularly in first-degree relatives or individuals with early onset disease) [4,5].

Currently, approximately 10 million men are diagnosed with prostate cancer worldwide, of which approximately 700,000 have metastasized [6]. Although PCa is typically diagnosed at an early stage, the risk-benefit ratio of treatment remains uncertain. It is one of the most controversial areas in medicine, as current treatment modalities still carry a high incidence of complications [4,7]. Evidence from fields such as genetics and pathology has demonstrated that inflammation plays a significant role in the initiation, promotion, progression, invasion, and metastasis of prostate cancer. Additionally, dietary components, including fats, proteins, carbohydrates, vitamins (A, D, and E), and polyphenols, have been shown through animal studies to contribute to the development and progression of prostate cancer. However, the precise mechanisms remain to be fully elucidated and the individual role of each factor remains a subject of considerable debate [8,9].

An increasing body of evidence suggests that nutritional and inflammatory status have significant prognostic and diagnostic value in patients with malignant tumors [10,11]. Studies have shown that the Neutrophil-to-Lymphocyte Ratio (NLR) is not a reliable marker for differentiating malignant and benign prostate conditions. However, the NLR may be valuable for further stratification of patients with prostate cancer [12]. The Neutrophil Percentage-to-Albumin Ratio (NPAR) is a novel indicator reflecting systemic inflammation and immune-related diseases, proposed by Bernard et al.

Studies have reported its prognostic significance in patients with rectal cancer who undergo surgery following neoadjuvant chemoradiotherapy. According to early

studies, NPAR can help predict acute kidney injury, cardiogenic shock, myocardial infarction, and patient prognosis [13–17]. However, the relationship between NPAR and prostate cancer remains unclear and has not been specifically studied. Therefore, we conducted a cross-sectional study using data from the National Health and Nutrition Examination Survey (NHANES) conducted between 2001 and 2018 to explore the association between NPAR and prostate cancer. This study aimed to provide meaningful insights into improved screening and prevention strategies for prostate cancer.

## Materials and methods

### Study population

We utilized data from the National Health and Nutrition Examination Survey (NHANES), a biennial cross-sectional study designed to evaluate the health and nutritional status of the U.S. population through standardized interviews and physical examinations [10]. The study protocol was approved by the NCHS and CDC institutional review boards. All participants provided written informed consent, and the sample was selected using a complex, multistage probability sampling strategy.

In this study, we used data extracted from NHANES survey cycles conducted between 2001 and 2018. This study included 21,644 male participants aged ≥ 20 years who provided detailed information on their cancer, nutrition, and inflammation status. We excluded individuals who lacked data on education level, marital status, household income, BMI, history of diabetes, hypertension, or other relevant variables as well as those with abnormal serological test results [10]. After exclusion, a total of 16,481 patients were included in the final analysis. Additionally, we randomly collected routine blood and liver function test reports from 82 prostate cancer patients and 70 with benign prostatic hyperplasia of Asian descent within two weeks prior to their diagnosis and calculated their NPAR values for comparison and validation. The clinical data were derived from serological tests conducted between June 2020 and June 2023, with final data collection completed on January 7, 2025. During and after data collection, the authors had access to identifiable participant information.

### The evaluation of the NPAR

The derivation formula for NPAR is NPAR = NEUT/Alb (serum albumin in grams per deciliter)[16]. NEUT represents the neutrophil percentage (%), not the absolute count. NPAR was classified as low (< 11.85), moderate (≥ 11.85 and < 15.07), and high (≥ 15.07). These thresholds were derived from the tertiles of the NPAR distribution within our NHANES study population.

### Assessment of prostate cancer

The medical condition survey was designed to collect self-reported diagnoses of prostate cancer (PCa). Participants were asked whether a doctor or another healthcare provider had informed them of having cancer or malignant tumors. Those who answered "yes" were further questioned regarding the type of cancer they had experienced. PCa patients were classified as those who reported only having PCa (including both primary and isolated tumors). Respondents who answer "no," have a history of other malignancies, or report PCa alongside other types of cancer are categorized as non-PCa(10). The diagnoses for these 152 clinical patients were established based on our standard protocol. We rely on clinical pathological reports to diagnose and distinguish between prostate cancer and benign prostatic hyperplasia. Even when imaging examinations or PSA levels strongly suggest benign prostatic hyperplasia, postoperative pathology remains the "gold standard" for definitive diagnosis.

### Covariate information

In our study, we selected covariates associated with the disease. These covariates included age (continuous), gender(male or female), race (Mexican American, Other Hispanic, Non-Hispanic White, Non-Hispanic Black, and

Other Race—including Multi-Racial), education level (less than high school, high school or equivalent, and college or above), alcohol consumption (non-drinking, low to moderate drinking, heavy drinking), smoking history (never smoked, former smoker, current smoker), BMI (< 25 kg/m2, 25–30 kg/m2, and > 30 kg/m2), marital status (married/ living with partner, widowed/divorced/separated, never married), poverty-to-income ratio (PIR) (< 1.30, 1.30–3.50, and > 3.50), physical activity (yes or no), cardiovascular disease (CVD) (yes or no), hypertension (yes or no), and diabetes (yes or no). Hypertension and diabetes status were determined based on self-reported physician diagnoses, as commonly used in NHANES-based studies [10]. CVD was defined as congestive heart failure, coronary artery disease, heart attack, myocardial infarction, angina, or stroke. Physical activity was defined as engaging in vigorous or moderate recreational activities [18,19]. We collected complete covariate data without any missing values for all 152 clinical patients, including age, BMI, smoking history, alcohol consumption, hypertension, diabetes, and CVD.

## Statistical methods

The data were analyzed based on NCHS weighting. Participants were divided into two groups based on the presence or absence of prostate cancer (PCa), and baseline characteristics were compared between the groups. Continuous variables are presented as means (standard deviation, SD), while categorical variables are presented as frequencies and percentages. The NPAR was treated as a continuous and categorical variable. Wilcoxon and chi-square tests were used to compare the associations between baseline characteristics for continuous and categorical variables, respectively. We employed multivariable logistic regression models to explore the relationship between NPAR and PCa: Model 1, without adjustment for covariates; model 2, adjusted for age, race, BMI, education, PIR, marital status, physical activity, smoking status, alcohol consumption, hypertension, diabetes, CVD, and other relevant covariates.

Next, we used the XGBoost algorithm to evaluate the relative importance of multiple variables that influence PCa. We configured the XGBoost model with a learning rate of 0.05 to optimize the trade-off between model complexity and generalizability. To mitigate overfitting, we restricted the maximum tree depth to 6 and set the number of trees to 500 [10]. These parameters were selected to enhance model stability and predictive accuracy while preserving its ability to generalize to unseen data [20,21]. Subsequently, restricted cubic splines were plotted using logistic regression to visually illustrate the relationship between the NPAR and PCa. The NPAR values underwent natural logarithm (ln) transformation to satisfy the assumption of normality. The model with the smallest Akaike Information Criterion (AIC) value was selected to test for potential nonlinear relationships between NPAR and PCa. Based on the smooth curve, inflection points were calculated using a recursive algorithm.

To confirm the reliability of our findings, we conducted extensive sensitivity analysis. First, we performed an initial analysis by excluding individuals with other malignancies from the control group to investigate potential biases associated with these individuals. Second, we stratified NPAR into tertiles to examine trend effects. Finally, we conducted subgroup analyses of the covariates and the interaction between NPAR and the covariates, adjusting the models to control for the potential confounding factors mentioned above.

For further validation, we utilized clinical data from 82 prostate cancer patients and 70 with benign prostatic hyperplasia (BPH), specifically NPAR values derived from routine blood and liver function test reports collected within two weeks prior to diagnosis. A non-parametric Mann-Whitney U test (Wilcoxon rank-sum test) was performed to assess the differences between the two groups. The datasets were combined and ranked and U-statistics were calculated, with a smaller U-value selected as the final statistic. Additionally, we employed a diagnostic ROC prediction model to evaluate the diagnostic performance of the NPAR in predicting outcomes.

All statistical analyses were performed using R software (version 4.2.0) with relevant R packages. In our study, a p-value < 0.05 was considered statistically significant.

## Ethics statement

**Informed consent for publication.** Consent for publication of raw data not obtained but dataset is fully anonymous in a manner that can easily be verified by any user of the dataset. Publication of the dataset clearly and obviously presents minimal risk to confidentiality of study participants. The retrospective study was conducted without obtaining explicit consent for data sharing or publication, making it impractical to obtain informed consent afterwards due to reasons such as patient death or loss of contact.

The analysis of the publicly available NHANES data did not require additional informed consent for publication, as all data are anonymized, and the National Center for Health Statistics (NCHS) Research Ethics Review Board approved the NHANES protocol with participant consent covering the use of data for research and publication.

## Results

### Baseline characteristics

A total of 16,481 participants were included, and their characteristics are presented according to PCa status (Table 1). Compared to the non-PCa group, we found that older age, non-Hispanic White and Black race, poverty-to-income ratio (PIR) of [1.30, 3.50], being married or living with a partner, former smoking status, hypertension, cardiovascular disease (CVD), non-drinking or heavy drinking habits, and non-diabetic individuals were more likely to develop PCa ($p < 0.05$). There was a significant difference in NPAR levels between PCa patients [median (IQR): 14.4 (2.75)] and the non-case group [median (IQR): 13.4 (2.57)]($p < 0.001$).

### The association between NPAR levels and the probability of prostate cancer

The association between NPAR levels and PCa prevalence was analyzed using weighted multivariable logistic regression, with NPAR levels treated as a categorical variable [10]. The results are presented in Table 2. We observed a positive correlation between PCa and NPAR levels within a specific range. Sensitivity analysis was conducted using NPAR tertiles, with tertile 1 (T1) as the reference group. In Model 1, the odds ratios (ORs) for T2 and T3 were 1.87 [95% CI: 1.24, 2.82] and 4.28 [95% CI: 2.70, 6.79], respectively, with a trend effect p-value < 0.002. In Model 2, the ORs for T2 and T3 were 1.48 [95% CI: 0.97, 2.26] and 1.94 [95% CI: 1.19, 3.18], respectively, with a trend effect p-value = 0.023. As the NPAR value increases, the OR value also increases.

In this study, the XGBoost(eXtreme Gradient Boosting) method was employed to construct a machine-learning model to evaluate the relative importance of the included variables. The XGBoost model demonstrated strong predictive performance, with a Mean Squared Error (MSE) of 0.11 and a Root Mean Squared Error (RMSE) of 0.32, indicating a close alignment between predicted and observed values and a high level of accuracy in the model's predictions [10]. Age, NPAR, race, smoking status, and education were identified as the five most relevant variables, further confirming that NPAR is one of the most significant factors influencing prostate cancer (PCa) prevalence (Fig 1).

### Exploring nonlinear relationships

Using Generalized Additive Models (GAM) and smooth curve fitting (penalized spline method) [10], we identified a nonlinear association between NPAR levels and the probability of PCa (Fig 2). In the fully adjusted model, a nonlinear relationship was observed between ln-transformed NPAR levels and PCa prevalence (p for nonlinearity = 0.0001). This finding suggests that the two-part regression model was appropriate for data fitting. A threshold of 13.46 for NPAR levels was determined, with 2.60 (ln transformation) identified as the inflection point using recursive methods and a two-segment regression model. When NPAR values exceeded 13.46, the odds ratio for PCa significantly increased, indicating that this NPAR threshold may serve as a potential preventive indicator for PCa. The relationship between NPAR and PCa was nonlinear and exhibited a threshold effect.

**Table 1. Baseline characteristics of participants stratified by prostate cancer (PCA) status.**

| | [ALL] | Control | PCa | p.overall |
|---|---|---|---|---|
| | *N=16481* | *N=16061* | *N=420* | |
| **Age, n (%)** | | | | <0.001 |
| <60 | 10933 (66.3%) | 10908 (67.9%) | 25 (5.95%) | |
| >=60 | 5548 (33.7%) | 5153 (32.1%) | 395 (94.0%) | |
| **Race, n (%)** | | | | <0.001 |
| Mexican American | 2787 (16.9%) | 2768 (17.2%) | 19 (4.52%) | |
| Other Hispanic | 1224 (7.43%) | 1199 (7.47%) | 25 (5.95%) | |
| Non-Hispanic White | 7915 (48.0%) | 7684 (47.8%) | 231 (55.0%) | |
| Non-Hispanic Black | 3166 (19.2%) | 3040 (18.9%) | 126 (30.0%) | |
| Other Race – Including Multi-Racial | 1389 (8.43%) | 1370 (8.53%) | 19 (4.52%) | |
| **BMI, n (%)** | | | | 0.461 |
| Normal weight | 4468 (27.1%) | 4365 (27.2%) | 103 (24.5%) | |
| Overweight | 6497 (39.4%) | 6323 (39.4%) | 174 (41.4%) | |
| Obese | 5516 (33.5%) | 5373 (33.5%) | 143 (34.0%) | |
| **Education, n (%)** | | | | 0.222 |
| Less than high school | 4185 (25.4%) | 4092 (25.5%) | 93 (22.1%) | |
| High school or equivalent | 4000 (24.3%) | 3900 (24.3%) | 100 (23.8%) | |
| College or above | 8296 (50.3%) | 8069 (50.2%) | 227 (54.0%) | |
| **Poverty-to-income ratio, n (%)** | | | | <0.001 |
| <1.30 | 4556 (27.6%) | 4479 (27.9%) | 77 (18.3%) | |
| 1.30–3.50 | 6322 (38.4%) | 6129 (38.2%) | 193 (46.0%) | |
| >3.50 | 5603 (34.0%) | 5453 (34.0%) | 150 (35.7%) | |
| **Marital status, n (%)** | | | | <0.001 |
| Married/Living with partner | 10995 (66.7%) | 10691 (66.6%) | 304 (72.4%) | |
| Widowed/Divorced/Separated | 2616 (15.9%) | 2513 (15.6%) | 103 (24.5%) | |
| Never married | 2870 (17.4%) | 2857 (17.8%) | 13 (3.10%) | |
| **Physical activity, n (%)** | | | | 0.336 |
| NO | 7448 (45.2%) | 7248 (45.1%) | 200 (47.6%) | |
| YES | 9033 (54.8%) | 8813 (54.9%) | 220 (52.4%) | |
| **Smoking status, n (%)** | | | | <0.001 |
| Never smoked | 6859 (41.6%) | 6707 (41.8%) | 152 (36.2%) | |
| Former smoker | 5350 (32.5%) | 5122 (31.9%) | 228 (54.3%) | |
| Current smoker | 4272 (25.9%) | 4232 (26.3%) | 40 (9.52%) | |
| **Alcohol drinking, n (%)** | | | | <0.001 |
| Non-drinking | 3483 (21.1%) | 3351 (20.9%) | 132 (31.4%) | |
| Low to moderate drinking | 12886 (78.2%) | 12602 (78.5%) | 284 (67.6%) | |
| Heavy drinking | 112 (0.68%) | 108 (0.67%) | 4 (0.95%) | |
| **Hypertension, n (%)** | | | | <0.001 |
| YES | 5691 (34.5%) | 5422 (33.8%) | 269 (64.0%) | |
| NO | 10790 (65.5%) | 10639 (66.2%) | 151 (36.0%) | |
| **Diabetes, n (%)** | | | | <0.001 |
| YES | 2159 (13.1%) | 2061 (12.8%) | 98 (23.3%) | |
| NO | 14322 (86.9%) | 14000 (87.2%) | 322 (76.7%) | |
| **CVD, n (%)** | | | | <0.001 |
| YES | 2099 (12.7%) | 1980 (12.3%) | 119 (28.3%) | |
| NO | 14382 (87.3%) | 14081 (87.7%) | 301 (71.7%) | |

*(Continued)*

**Table 1.** (Continued)

|  | [ALL] | Control | PCa | p.overall |
|---|---|---|---|---|
|  | N=16481 | N=16061 | N=420 |  |
| **NPAR** | 13.4 (2.58) | 13.4 (2.57) | 14.4 (2.75) | <0.001 |

Continuous variables were analyzed using the Wilcoxon test and are presented as median (IQR), and the chi-square test was used to analyze categorical variables, presented as N (%).

Abbreviations: CVD, cardiovascular disease; BMI, body mass index.

**Table 2. Weighted multivariable logistic regression.**

| NPAR | Control(n=16061) | Prostatic Cancer(n=420) | Model 1 | P-value | Model 2 | P-value |
|---|---|---|---|---|---|---|
| Low | 4051 (25.2%) | 64 (15.2%) | Ref |  | Ref |  |
| Intermediate | 8042 (50.1%) | 190 (45.2%) | 1.87[1.24,2.82] | 0.03 | 1.48[0.97,2.26] | 0.066 |
| High | 3968 (24.7%) | 166 (39.5%) | 4.28[2.70,6.79] | <0.001 | 1.94[1.19,3.18] | 0.009 |
| P for trend |  |  |  | <0.001 |  | 0.023 |

Model 1 was unadjusted.

Model 2 was adjusted for age, race, BMI, education, PIR, marital status, physical activity, smoking status, alcohol consumption, hypertension, diabetes, and CVD.

## Sensitivity analysis

A series of sensitivity analyses were conducted to assess the robustness of the main findings [10]. First, NPAR was treated as a categorical variable using quartiles to examine end effects (Table 2). Further subgroup analyses demonstrated that the association between NPAR and PCa remained consistent (Fig 3). The stratified association between NPAR levels and the incidence of PCa was examined. In this model, all covariates were adjusted for, except the stratification variables themselves. (Abbreviations: BMI, body mass index; PIR, poverty-income ratio; CVD, cardiovascular disease; OR, odds ratio; CI, confidence interval) Significant interactions were observed between factors, such as age, race, and smoking status. These interactions suggest that the relationship between NPAR levels and PCa prevalence varies across the subgroups.

## Clinical validation

To illustrate the distribution of NPAR values in 82 patients with prostate cancer (I) and 70 patients with BPH(J), a non-parametric Mann-Whitney U test (Wilcoxon rank-sum test) was used to assess the differences [Fig 4]. The mean NPAR values for the two groups were 17.47 and 14.84, respectively. The rank difference between the two groups was −2.7857 [−3.7675, −1.6251](J-I), with a test statistic of 4058.5 and p-value<0.001.

The reliability of the results was assessed using a diagnostic ROC curve prediction model [Fig 5]. The outcome was inversely predicted with a model cut-off value of 15.964 and an AUC of 0.707 [0.623, 0.791], indicating that the model has a moderate level of accuracy.

The same statistical and variable categorization methods as in Table 1 were employed. Their characteristics are presented according to diagnosis (Table 3). Compared to the BPH group, we found that former smoking status and never smoked individuals were more likely to develop PCa (p<0.05). There was a significant difference in NPAR levels between PCa patients [median (IQR): 17.5 (4.21)] and the BPH group [median (IQR): 14.8 (3.52)](p<0.001).

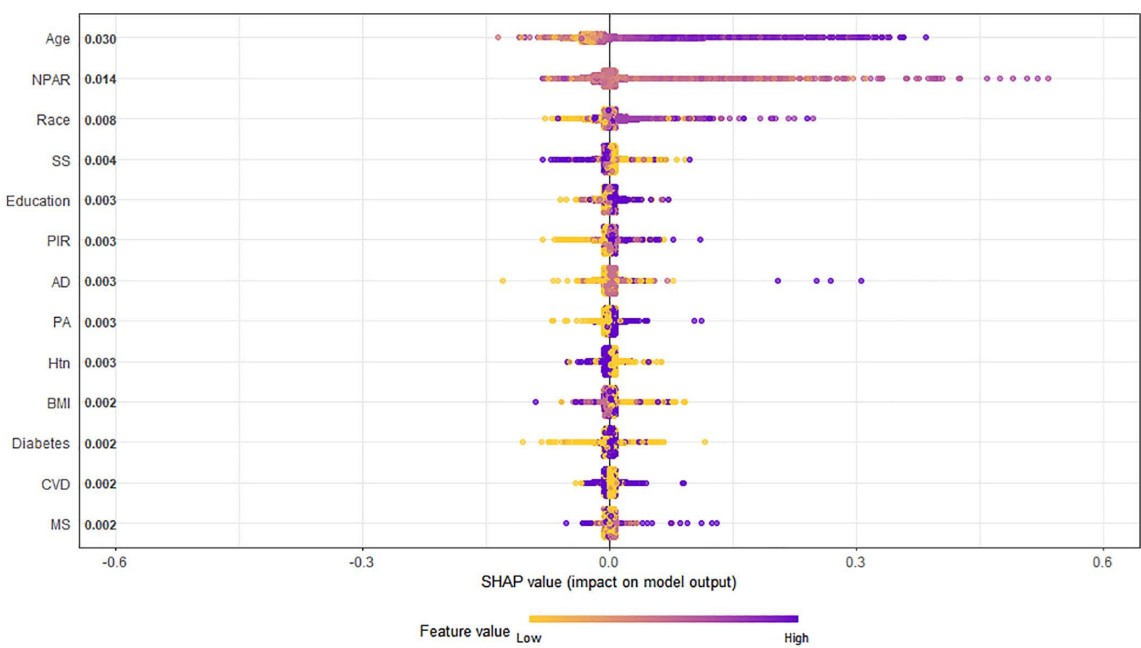

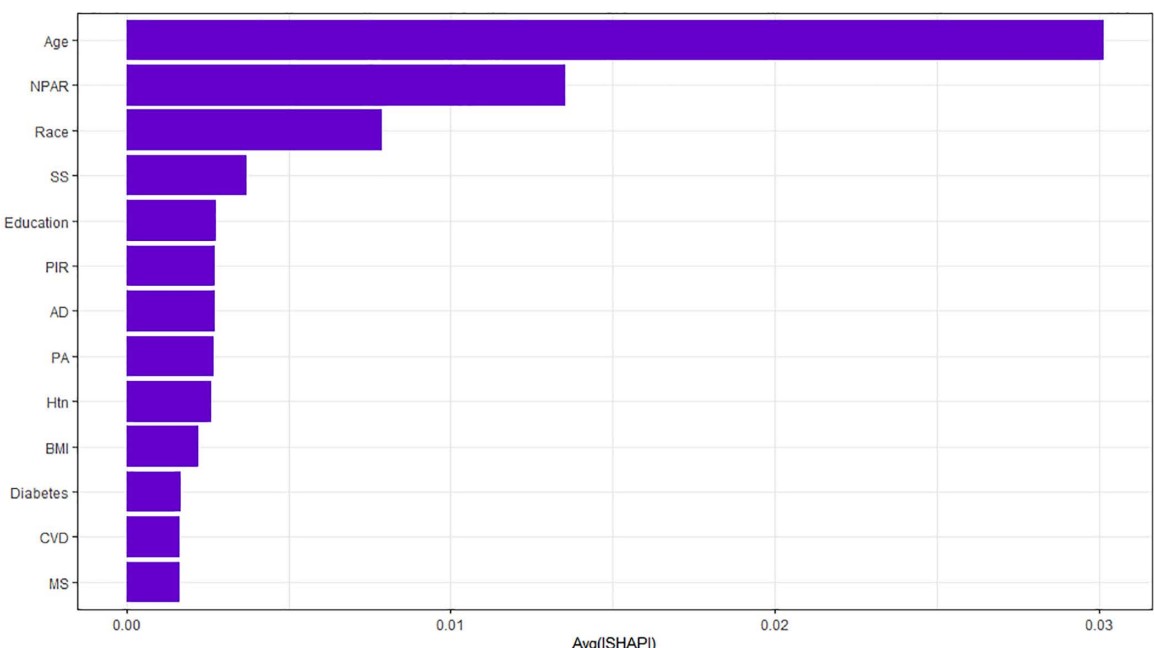

**Fig 1. Machine-learning model.** (A)(B) SHAP provides a natural explanation framework for XGBoost by assigning feature contribution values, which helps us understand the model's decision-making process. Credit: The authors. (Abbreviations: SS, smoking status; PIR, Poverty-to-income ratio; AD, Alcohol consumption; Htn, Hypertension; MS, Marital status; PA,Physical activity).

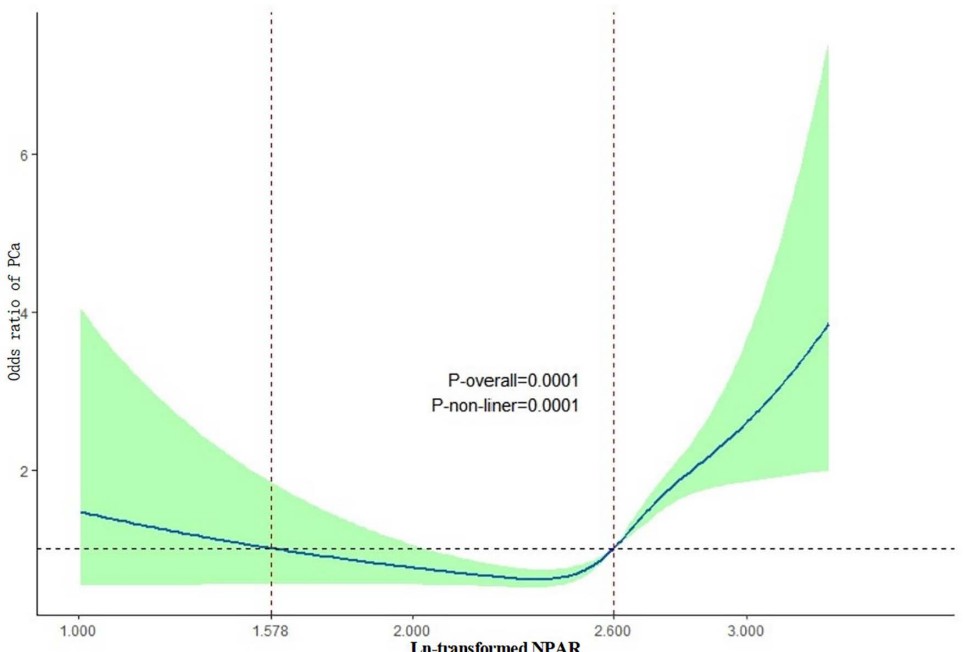

**Fig 2. Generalized additive model.** Based on the odds ratio for PCa at the ln-transformed NPAR levels in the general population, the solid line and shaded area represent the odds ratio for PCa and its 95% confidence interval, respectively. The vertical dashed line indicates the threshold for the lowest PCa risk (ln-transformed NPAR = 2.60). All covariates were adjusted for in this model. Credit: The authors.

## Discussion

Based on our findings, this study explored the relationship between NPAR values and prostate cancer (PCa) diagnosis. Through multivariable regression analysis and restricted cubic spline fitting using Generalized Additive Models (GAM), the study validated a positive linear correlation between NPAR levels and PCa incidence. In addition, the machine learning model further highlighted the significance of NPAR in PCa diagnosis. Clinical validation studies also suggest that NPAR has potential clinical diagnostic value. Sensitivity analyses demonstrated consistent results across different analytical approaches.

Existing studies have demonstrated a significant association between systemic inflammatory responses and carcinogenic effects [22,23]. Cytokine release induced by chronic inflammation can lead to genetic mutations, alterations in the expression of oncogenes and tumor suppressor genes, inhibition of apoptosis, and promotion of angiogenesis, resulting in dysregulation of inflammatory signaling pathways. Additionally, chronic inflammation can foster the formation of an immunosuppressive tumor microenvironment (TME) by recruiting various immunosuppressive cells, such as M2 tumor-associated macrophages (M2-TAMs), myeloid-derived suppressor cells (MDSCs), and regulatory T cells (Tregs), thereby promoting tumor initiation and progression. Systemic chronic inflammation, such as that associated with obesity,depression, and treatment-induced chronic inflammation, may also contribute to tumor development by disrupting immune system homeostasis [24].Elevated neutrophil counts can promote cancer progression and metastasis through multiple mechanisms, including stimulation of angiogenesis and impairment of T-cell-dependent antitumor immunity [25,26].

Albumin level is commonly used as an indicator for nutritional assessment. Albumin exhibits anti-inflammatory properties. Compared to the low albumin group, the high albumin group showed a significant reduction in pro-inflammatory

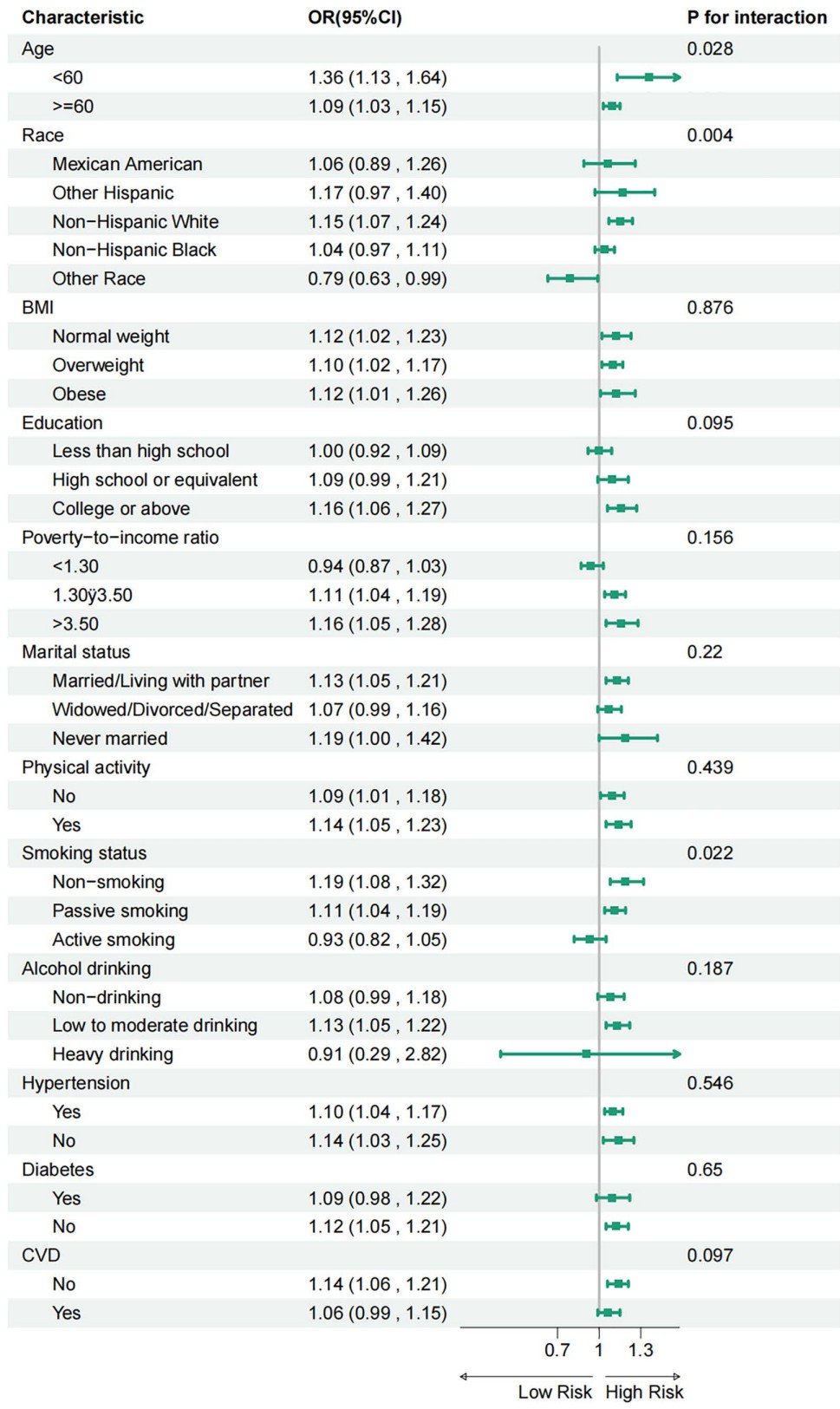

| Characteristic | OR(95%CI) | P for interaction |
|---|---|---|
| Age | | 0.028 |
| <60 | 1.36 (1.13 , 1.64) | |
| >=60 | 1.09 (1.03 , 1.15) | |
| Race | | 0.004 |
| Mexican American | 1.06 (0.89 , 1.26) | |
| Other Hispanic | 1.17 (0.97 , 1.40) | |
| Non−Hispanic White | 1.15 (1.07 , 1.24) | |
| Non−Hispanic Black | 1.04 (0.97 , 1.11) | |
| Other Race | 0.79 (0.63 , 0.99) | |
| BMI | | 0.876 |
| Normal weight | 1.12 (1.02 , 1.23) | |
| Overweight | 1.10 (1.02 , 1.17) | |
| Obese | 1.12 (1.01 , 1.26) | |
| Education | | 0.095 |
| Less than high school | 1.00 (0.92 , 1.09) | |
| High school or equivalent | 1.09 (0.99 , 1.21) | |
| College or above | 1.16 (1.06 , 1.27) | |
| Poverty−to−income ratio | | 0.156 |
| <1.30 | 0.94 (0.87 , 1.03) | |
| 1.30ÿ3.50 | 1.11 (1.04 , 1.19) | |
| >3.50 | 1.16 (1.05 , 1.28) | |
| Marital status | | 0.22 |
| Married/Living with partner | 1.13 (1.05 , 1.21) | |
| Widowed/Divorced/Separated | 1.07 (0.99 , 1.16) | |
| Never married | 1.19 (1.00 , 1.42) | |
| Physical activity | | 0.439 |
| No | 1.09 (1.01 , 1.18) | |
| Yes | 1.14 (1.05 , 1.23) | |
| Smoking status | | 0.022 |
| Non−smoking | 1.19 (1.08 , 1.32) | |
| Passive smoking | 1.11 (1.04 , 1.19) | |
| Active smoking | 0.93 (0.82 , 1.05) | |
| Alcohol drinking | | 0.187 |
| Non−drinking | 1.08 (0.99 , 1.18) | |
| Low to moderate drinking | 1.13 (1.05 , 1.22) | |
| Heavy drinking | 0.91 (0.29 , 2.82) | |
| Hypertension | | 0.546 |
| Yes | 1.10 (1.04 , 1.17) | |
| No | 1.14 (1.03 , 1.25) | |
| Diabetes | | 0.65 |
| Yes | 1.09 (0.98 , 1.22) | |
| No | 1.12 (1.05 , 1.21) | |
| CVD | | 0.097 |
| No | 1.14 (1.06 , 1.21) | |
| Yes | 1.06 (0.99 , 1.15) | |

0.7  1  1.3

← Low Risk   High Risk →

**Fig 3. Subgroup analysis.**

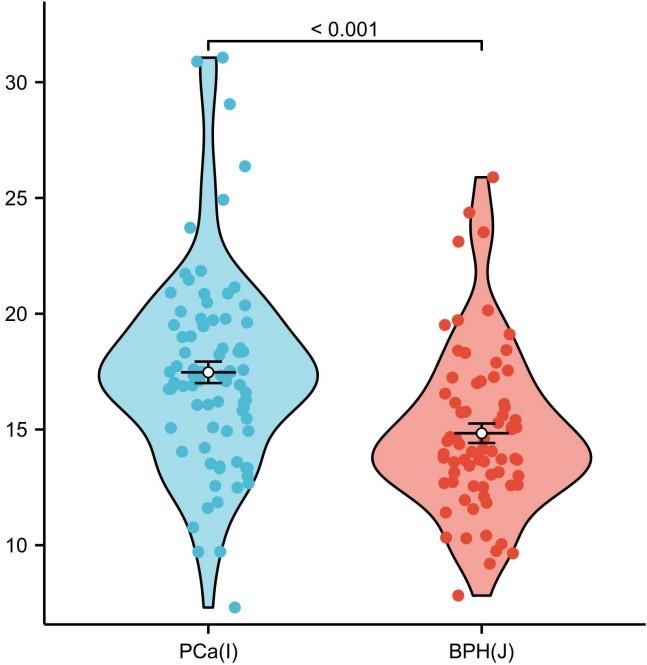

**Fig 4. Group comparison.**

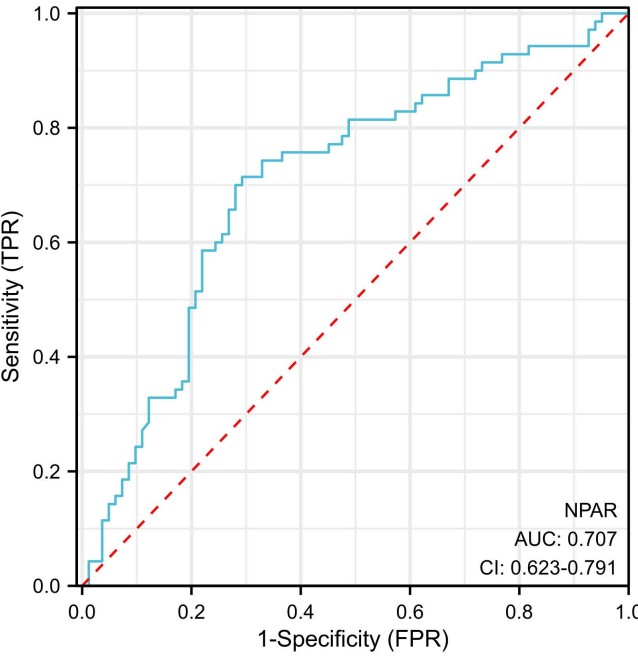

**Fig 5. Diagnostic ROC curve.** False Positive Rate (FPR) is the proportion of negative class samples incorrectly classified as positive by the model, which is also known as the false alarm rate. True Positive Rate (TPR) is the proportion of positive class samples correctly classified by the model, also known as recall. Credit: The authors.

**Table 3. Baseline characteristics by prostate cancer (PCa) and benign prostatic hyperplasia (BPH) status.**

|  | [ALL] | BPH | PCa | p.overall |
|---|---|---|---|---|
|  | *N=152* | *N=70* | *N=82* |  |
| **Age**, n(%) |  |  |  | 0.266 |
| <60 | 18 (11.8%) | 11 (15.7%) | 7 (8.54%) |  |
| >=60 | 134 (88.2%) | 59 (84.3%) | 75 (91.5%) |  |
| **BMI**, n(%) |  |  |  | 0.629 |
| Normal weight | 112 (73.7%) | 52 (74.3%) | 60 (73.2%) |  |
| Overweight | 38 (25.0%) | 18 (25.7%) | 20 (24.4%) |  |
| Obese | 2 (1.32%) | 0 (0.00%) | 2 (2.44%) |  |
| **Smoking status**, n (%) |  |  |  | 0.046 |
| Never smoked | 85 (55.9%) | 33 (47.1%) | 52 (63.4%) |  |
| Former smoker | 27 (17.8%) | 12 (17.1%) | 15 (18.3%) |  |
| Current smoker | 40 (26.3%) | 25 (35.7%) | 15 (18.3%) |  |
| **Alcohol drinking**, n (%) |  |  |  | 0.938 |
| Non-drinking | 102 (67.1%) | 46 (65.7%) | 56 (68.3%) |  |
| Low to moderate drinking | 35 (23.0%) | 17 (24.3%) | 18 (22.0%) |  |
| Heavy drinking | 15 (9.87%) | 7 (10.0%) | 8 (9.76%) |  |
| **Hypertension**, n (%) |  |  |  | 0.523 |
| YES | 77 (50.7%) | 33 (47.1%) | 44 (53.7%) |  |
| NO | 75 (49.3%) | 37 (52.9%) | 38 (46.3%) |  |
| **Diabetes**, n (%) |  |  |  | 0.450 |
| YES | 22 (14.5%) | 8 (11.4%) | 14 (17.1%) |  |
| NO | 130 (85.5%) | 62 (88.6%) | 68 (82.9%) |  |
| **CVD**, n (%) |  |  |  | 0.262 |
| YES | 31 (20.4%) | 11 (15.7%) | 20 (24.4%) |  |
| NO | 121 (79.6%) | 59 (84.3%) | 62 (75.6%) |  |
| **NPAR** | 16.3 (4.11) | 14.8 (3.52) | 17.5 (4.21) | <0.001 |

Abbreviations: CVD, cardiovascular disease; BMI, body mass index; BPH, benign prostatic hyperplasia.

cytokines,such as TNF and CRP levels [27]. Albumin is an important medium-sized carrier protein that plays a vital role in various physiological functions, including osmotic regulation, antioxidation, anti-inflammatory actions, nutrient and drug transport, and acid-base balance regulation. It constitutes more than half of the total serum protein level. Diseases such as liver cancer and cirrhosis not only reduce albumin synthesis, but also alter its structure and function, leading to complications associated with impaired physiological functions. These changes further compromise the body's ability to maintain homeostasis and perform essential processes. Reduced albumin levels and altered functionality are commonly used as clinical markers to assess disease severity and prognosis in liver conditions [28–30]. Neutrophils and albumin play crucial roles in inflammation and immune responses. Recent studies have linked these effects to the efficacy, prevention, and prognosis of cancer treatment [31–35]. Neutrophils and albumin play crucial roles in inflammation and immune responses. Recent studies have linked these effects to the efficacy, prevention, and prognosis of cancer treatment [36]. During tumor progression, elevated neutrophil levels and increased tumor burden lead to a reduction in albumin levels in the body.

NPAR serves as a composite indicator of two key pathophysiological states: neutrophilia (indicative of ongoing inflammation) and hypoalbuminemia (indicative of nutritional deficiency and chronic inflammatory consumption). A high neutrophil count fosters a pro-tumorigenic milieu by releasing reactive oxygen species (ROS) and proteases like matrix metalloproteinase-9 (MMP-9), which can cause DNA damage in prostate epithelial cells and facilitate tissue remodeling

for invasion and metastasis [25,26]. Concurrently, the systemic inflammatory response stimulates the release of cytokines such as IL-6, TNF-α, and IL-1β, which not only further promote neutrophil recruitment but also suppress albumin synthesis by the liver [27,28]. This creates a vicious cycle amplifying both components of the NPAR. Within the prostate TME, neutrophils are polarized towards a pro-tumor (N2) phenotype. These tumor-associated neutrophils (TANs) contribute to immune evasion by inhibiting the cytotoxic activity of CD8 + T cells and natural killer (NK) cells, and by recruiting other immunosuppressive cells, including M2 macrophages and regulatory T cells (Tregs) [24,25]. Albumin, beyond its nutritional role, possesses significant anti-inflammatory and antioxidant properties. Low albumin levels in the TME may thus diminish these protective effects, allowing inflammation to proceed unchecked and further exacerbating immunosuppression [27–30]. Therefore, NPAR potentially reflects the degree of this immunosuppressive, pro-inflammatory shift in the TME.

The NPAR thresholds used in this study (low: < 11.85; moderate: 11.85–15.07; high: ≥ 15.07), we acknowledge that these thresholds are derived from a U.S.-based cohort and may not be directly generalizable to other ethnic or geographic populations. In our Asian validation cohort, the optimal diagnostic cutoff for NPAR was found to be 15.96, which is higher than the high-risk threshold in the NHANES cohort. This suggests that NPAR thresholds may vary by population due to differences in genetic background, lifestyle, or baseline health status.

This study has several strengths. First, this is the first national study to investigate the direct association between NPAR levels in males and PCa, providing novel insights into the diagnostic value of nutritional and inflammatory status in PCa. Second, the use of machine learning models underscores the importance of the NPAR in predicting PCa. Compared with traditional statistical methods, machine learning can handle large numbers of variables and complex relationships more effectively, thereby enhancing the accuracy and reliability of predictions. Additionally, extensive sensitivity analyses were conducted to confirm the robustness of the findings. Furthermore, NPAR demonstrates superior discriminative ability and accuracy compared to other nutritional and inflammatory biomarkers, suggesting its potential as a novel biomarker for early screening and diagnosis of PCa. Future studies could further explore the correlation between NPAR and PCa progression as well as its potential clinical applications. Finally, we validated our findings using clinical data, considering the potential liver dysfunction in patients with prostate cancer undergoing endocrine therapy, which may result in hypoalbuminemia as a side effect [37]. Therefore, we collected serological test data from patients before definitive diagnosis.

However, our study had several limitations. First, since the data were obtained from the NHANES database, the assessment of PCa relies on self-reported questionnaire responses and lacks precise clinical diagnostic indicators. This may have introduced classification bias in the estimation of PCa prevalence among the participants. Participants with undiagnosed or asymptomatic PCa might have been erroneously included in the non-PCa group, while those who misreported other conditions (e.g., benign prostatic hyperplasia) as cancer could have been included in the PCa group. This non-differential misclassification (as the error in reporting is likely unrelated to their NPAR levels) would most likely attenuate the observed association towards the null, meaning that the true odds ratio between NPAR and PCa might be stronger than what we have reported. This study did not conduct a sensitivity analysis to account for this potential bias; this limitation awaits further investigation in future research. Second, due to the cross-sectional nature of the NHANES data, our study can only demonstrate an association between NPAR and PCa prevalence, but cannot establish a causal relationship. The cross-sectional design prohibits inference of causality. Temporal sequence between NPAR levels and PCa diagnosis remains unclear. Third, although we attempted to control for as many covariates as possible to mitigate confounding biases, potential confounders, such as socioeconomic status, environmental exposure, and important clinical factors including PSA screening frequency, medication (e.g., statins, anti-inflammatories) use and family history, were either not measured or not included in the study, which may have affected the results. Although PSA is a well-established biomarker for PCa screening, it was not included in our analysis due to substantial missing data in the NHANES database. Additionally, PSA levels may be influenced by benign prostatic conditions (e.g., BPH) and inflammation, which could introduce confounding. Future studies with complete PSA data are warranted to validate the independent predictive value of NPAR alongside PSA.

## Conclusion

A positive correlation was observed between NPAR values and the prevalence of prostate cancer; however, the cross-sectional study design precludes the establishment of a causal relationship. This association is clinically significant, with NPAR cutoff values of 13.46 in the NHANES cohort and 15.96 in the Asian validation cohort. Elevated NPAR levels exhibit a discriminative capacity for the detection of PCa and may serve as a potential complementary marker.

## Acknowledgments

The authors thank all the participants and data contributors.

## Author contributions

**Conceptualization:** Ping Kong, Lei Yang, Haibing Wang.

**Data curation:** Ping Kong, Lei Yang, Haibing Wang, Zhiqi Liu.

**Methodology:** Ping Kong, Lei Yang, Haibing Wang.

**Validation:** Lei Yang.

**Writing – original draft:** Ping Kong, Lei Yang, Haibing Wang.

**Writing – review & editing:** Zhiqiang Zhang.

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
