## [Decision Letter · Decision Letter 0]

3 Sep 2025

Dear Dr. Zhang,

Thank you for submitting your manuscript to PLOS ONE. After careful consideration, we feel that it has merit but does not fully meet PLOS ONE’s publication criteria as it currently stands. Therefore, we invite you to submit a revised version of the manuscript that addresses the points raised during the review process.

This manuscript presents a cross-sectional study investigating the association between the Neutrophil Percentage-to-Albumin Ratio (NPAR) and the prevalence of prostate cancer (PCa). The authors utilize a large dataset from the National Health and Nutrition Examination Survey (NHANES) from 2001-2018, including 16,481 male participants. The study employs robust statistical methods, including multivariable logistic regression and an XGBoost machine learning model, to analyze the relationship. A key finding is a positive, nonlinear association between NPAR and PCa, with a specific threshold effect identified. The study's strengths include its novelty as the first to explore this specific biomarker for PCa in a large national cohort, the use of advanced statistical modeling, and the inclusion of a separate clinical cohort for validation.Overall, the research question is clinically relevant, and the study is well-structured. However, there are several major concerns regarding the study's design limitations and reporting that need to be addressed before it can be considered for publication. ;

We look forward to receiving your revised manuscript.

Kind regards,

Rocco Simone Flammia

Academic Editor

PLOS ONE

Journal Requirements:

https://doi.org/10.1186/s12889-024-19727-9

In your revision ensure you cite all your sources (including your own works), and quote or rephrase any duplicated text outside the methods section. Further consideration is dependent on these concerns being addressed.

“Precision Diagnosis and Classification of Prostate Cancer via Artificial Neural Network-Driven Intelligent Analysis of MR Images”

4. Thank you for stating the following in your manuscript: 

“This research was funded by the Research on Precision Diagnosis and Classification of Prostate Cancer Based on Artificial Neural Network-Driven Intelligent Analysis of MR Images(grant number 2023AH053161).”

“Precision Diagnosis and Classification of Prostate Cancer via Artificial Neural Network-Driven Intelligent Analysis of MR Images”

5. We note that you have indicated that there are restrictions to data sharing for this study. For studies involving human research participant data or other sensitive data, we encourage authors to share de-identified or anonymized data. However, when data cannot be publicly shared for ethical reasons, we allow authors to make their data sets available upon request. For information on unacceptable data access restrictions, please see http://journals.plos.org/plosone/s/data-availability#loc-unacceptable-data-access-restrictions. 

6. In this instance it seems there may be acceptable restrictions in place that prevent the public sharing of your minimal data. However, in line with our goal of ensuring long-term data availability to all interested researchers, PLOS’ Data Policy states that authors cannot be the sole named individuals responsible for ensuring data access (http://journals.plos.org/plosone/s/data-availability#loc-acceptable-data-sharing-methods).

7. PLOS requires an ORCID iD for the corresponding author in Editorial Manager on papers submitted after December 6th, 2016. Please ensure that you have an ORCID iD and that it is validated in Editorial Manager. To do this, go to ‘Update my Information’ (in the upper left-hand corner of the main menu), and click on the Fetch/Validate link next to the ORCID field. This will take you to the ORCID site and allow you to create a new iD or authenticate a pre-existing iD in Editorial Manager.

8. Your ethics statement should only appear in the Methods section of your manuscript. If your ethics statement is written in any section besides the Methods, please move it to the Methods section and delete it from any other section. Please ensure that your ethics statement is included in your manuscript, as the ethics statement entered into the online submission form will not be published alongside your manuscript. 

**Additional Editor Comments:**

Major Points for Revision

Limitations of Cross-Sectional Design and Self-Reported Data: The authors correctly acknowledge that the cross-sectional design prevents establishing causality and that the reliance on self-reported PCa diagnoses is a limitation. However, the impact of these limitations needs to be more strongly emphasized throughout the manuscript, particularly in the Abstract and Discussion.

Causality: Language implying causality should be carefully revised. For example, stating NPAR is positively correlated with the "risk" of PCa can be misleading in a cross-sectional context, which measures prevalence, not incidence. "Association with PCa prevalence" would be more accurate.

Misclassification Bias: Self-reported cancer diagnoses are prone to significant misclassification bias. The Discussion should elaborate on how this might have affected the results (e.g., potentially under- or overestimating the association's strength). Did the authors consider any sensitivity analysis to account for this potential bias?

Insufficient Detail on the Clinical Validation Cohort: While the inclusion of a validation cohort is a major strength, the description is insufficient.

The manuscript reports NPAR values for 82 PCa patients and 72 with BPH. Please provide a baseline characteristics table for this cohort (similar to Table 1) including age, race, and other relevant comorbidities. Without this, it's impossible to determine if the observed differences in NPAR are due to PCa or other confounding factors (e.g., an age mismatch between groups).

Please provide more context on the clinical diagnoses. For the PCa group, were data on Gleason score, tumor stage, or PSA levels available? For the BPH group, how was PCa definitively ruled out?

Lack of Critical Clinical Covariates: The analysis of NHANES data lacks key clinical variables for prostate cancer, most notably Prostate-Specific Antigen (PSA) levels. While NHANES may not collect detailed cancer staging information, PSA data is often available. Its omission is a significant gap. The authors should clarify if PSA data were available and, if so, why they were not included in the model. If unavailable, this should be stated as a major limitation.

Minor Points for Revision

Improper Citation: In the reference list, reference [8] is listed as "<inflammation and prostate cancer.pdf>". This is not a valid citation and must be replaced with the full, properly formatted reference.

Ambiguous Terminology: The term "NPAR" is defined as NEUT/Alb. The title and abstract specify "neutrophil percentage," but the abbreviation "NEUT" is ambiguous. Please clarify explicitly in the Methods section that NEUT represents the neutrophil percentage (%), not the absolute count, to avoid confusion.

Vague Language: The conclusion in the abstract states, "To some extent, NPAR is positively correlated with the risk of PCa...". The phrase "To some extent" is imprecise and weakens the statement. Please rephrase this for clarity and confidence (e.g., "A positive correlation was observed between NPAR and the prevalence of PCa...").

Figure Clarity: In the text describing the clinical validation, the authors refer to PCa patients as group (I) and BPH patients as group (J). These labels do not appear in the corresponding Figure 4. Please ensure all figures are clearly and accurately labeled to match the text.

Reviewers' comments:

Reviewer's Responses to Questions

1. Is the manuscript technically sound, and do the data support the conclusions?

Reviewer #1: Yes

2. Has the statistical analysis been performed appropriately and rigorously?

Reviewer #1: Yes

3. Have the authors made all data underlying the findings in their manuscript fully available?

Reviewer #1: Yes

4. Is the manuscript presented in an intelligible fashion and written in standard English?

Reviewer #1: Yes

Reviewer #1: Dear Authors,

Thank you for submitting your manuscript titled "The relationship between the neutrophil percentage to albumin ratio and the occurrence of prostate cancer" (Manuscript ID: PONE-D-25-16815).

Concerns:

•PCa status was self-reported, introducing potential bias in classification. Please acknowledge this more directly in the limitations.

•The cross-sectional design prohibits inference of causality. Emphasize this more clearly in your discussion and conclusion.

•Important clinical factors such as PSA screening frequency and medication (e.g., statins, anti-inflammatories), and family history use are not accounted for. Note this as a limitation.

•Clarify the thresholds used for categorizing NPAR and their applicability across diverse populations.

•The reported AUC of 0.707 suggests moderate diagnostic performance. Please reframe NPAR as a potential complementary marker rather than a standalone diagnostic marker.

•Biological Plausibility: Expand on the mechanistic rationale for why NPAR may be associated with prostate cancer risk, including its relation to inflammation and immune function in the tumor microenvironment.

I recommend a major revision to address these concerns. I look forward to receiving your revised manuscript.

Do you want your identity to be public for this peer review? For information about this choice, including consent withdrawal, please see our Privacy Policy

Reviewer #1: Yes: Akram Aldilaimi

---

## [Author Response · Author response to Decision Letter 1]

24 Sep 2025

11.PCa status was self-reported, introducing potential bias in classification. Please acknowledge this more directly in the limitations. We have already incorporated and emphasized this point in the limitations section.

12.The cross-sectional design prohibits inference of causality. Emphasize this more clearly in your discussion and conclusion. We have emphasized this more clearly in discussion and conclusion.

13.Important clinical factors such as PSA screening frequency and medication (e.g., statins, anti-inflammatories), and family history use are not accounted for. Note this as a limitation. This point has been incorporated into the limitations.

14.Clarify the thresholds used for categorizing NPAR and their applicability across diverse populations. We have refined both the Methods and Discussion sections.

15.Please reframe NPAR as a potential complementary marker rather than a standalone diagnostic marker. We have reframed it in Conclusion section.

16.Biological Plausibility: We have supplemented this point in the Discussion section.

We sincerely thank you for valuable comments.

---

## [Editor Report · Decision Letter 1]

28 Sep 2025

The relationship between the neutrophil percentage to albumin ratio and the occurrence of prostate cancer

PONE-D-25-16815R1

Dear Dr. Zhang,

We’re pleased to inform you that your manuscript has been judged scientifically suitable for publication and will be formally accepted for publication once it meets all outstanding technical requirements.

Kind regards,

Rocco Simone Flammia

Academic Editor

PLOS ONE

Additional Editor Comments (optional):

no

Reviewers' comments:

add paragraphs numbering

---

## [Editor Report · Acceptance letter]

PONE-D-25-16815R1

PLOS ONE

Dear Dr. Zhang,

I'm pleased to inform you that your manuscript has been deemed suitable for publication in PLOS ONE. Congratulations! Your manuscript is now being handed over to our production team.

Kind regards,

on behalf of

Dr. Rocco Simone Flammia

Academic Editor

PLOS ONE